# CONTRIBUTION OF INTERNAL REFLECTION IN LANGUAGE EMERGENCE WITH AN UNDER-RESTRICTED SITUATION

## ABSTRACT

Owing to language emergence, human beings have been able to understand the intentions of others, generate common concepts, and extend new concepts. Artificial intelligence researchers have not only predicted words and sentences statistically in machine learning, but also created a language system by communicating with the machine itself. However, strong constraints are exhibited in current studies (supervisor signals and rewards exist, or the concepts were fixed on only a point), thus hindering the emergence of real-world languages. In this study, we improved Batali (1998) and Choi et al. (2018)'s research and attempted language emergence under conditions of low constraints such as human language generation. We included the bias that exists in humans as an "internal reflection function" into the system. Irrespective of function, messages corresponding to the label could be generated. However, through qualitative and quantitative analysis, we confirmed that the internal reflection function caused "overlearning" and different structuring of message patterns. This result suggested that the internal reflection function performed effectively in creating a grounding language from raw images with an under-restricted situation such as human language generation.

## 1 INTRODUCTION

For artificial intelligence studies, it is important to build machines that can handle words and concepts (Bruner, 1981; Mikolov et al., 2016). Deep learning, especially the attention mechanism, has improved the success of various natural language processing tasks (Xu et al., 2018; Das et al., 2018; Mao et al., 2019; Das et al., 2017a; Huang et al., 2016). Furthermore, studies on multi-modal information are progressing. Supervised learning based on large datasets regards languages as statistical pattern processing with discrete symbols. Humans, as well as animals, have emerged information and concepts through the exchange of languages (Lupyan & Bergen, 2016; Lupyan & Clark, 2015). Languages enable the understanding of the intentions of others and a socially common system to be built (Bloom, 2002; Frank et al., 2009; Tomasello, 2010). In addition, languages can be used to recursively construct concepts and create new ones (Berwick & Chomsky, 2016; Hauser et al., 2002). Language intelligence involves more than statistically predicting words and sentences. It is important for the intelligence to formulate and systematize a language from contexts and situations.

Introducing the language system using the current datasets has been difficult in machine learning (Lake et al., 2017; Niven & Kao, 2019; Dua et al., 2019). Researchers have attempted to build agents to create the language system by communicating with each agent instead of language processing using statistical information (Steels, 2003; Mikolov et al., 2016; Rahwan et al., 2019). Complex tasks could be solved using these generated messages. Formal rules and messages that emerge owing to this communication have a system that differs from that of a human.

However, current language emergence studies have not achieved the autonomous grounding of mental images as those of humans and animals (Kottur et al., 2017; Bouchacourt & Baroni, 2018). Two issues remain: (1) The correspondence of generated messages to concepts is mediated by the architecture. Messages cannot be directly grounded to concepts. In some cases, a meaningless message was generated, and in other cases, a meaningful message was assigned to random noise images. (2) In current studies, the "supervisor" role is essential because agents are rewarded for acquiring the

language system when communication is established. The supervisor method provided implicit answers such as "direction of correct communication for task resolutions" and "direction of a correct language system corresponding to a concept." Few studies have focused on language systems that emerge from prior knowledge regarding pure machines.

These issues appear in language emergence models, which are different from the emergence and learning of human languages. Humans have prior knowledge of the concept and corresponding grounding messages (Imai & Masuda, 2013; Milligan et al., 2007). They can create language systems independently from the supervisor (Senghas et al., 2004). Little has been reported on language emergence under these conditions, in which the agent possesses individual prior knowledge and is independent of the supervisor.

The purpose of this study is to confirm a compositional language emergence method, in which machines autonomously ground concepts to each agent in an almost human-like manner. We developed a method and architecture that can generate a language from discrete symbols. In a human-like manner, agents must autonomously generate messages corresponding to cluster images that are not fixed at one point. To create formal rules within these under-restricted situations, we introduce the human cognitive bias (Brennan & Clark, 1996; Lake et al., 2019). Using the generated messages and the reconstructed images, we compared the characteristics of the agreed formal rules by bias and discovered the emergent parameters necessary for the rules.

## 2 RELATED WORK

Artificial life studies of language emergence have discovered the dynamism of the generated language and the condition under which language systematization and formality has emerged (Steels, 2000; Martin & White, 2003; Nolfi & Mirolli, 2009). Recently, a neural network model has been used to generate a complex language using reinforcement learning or supervised learning. It is classified into (1) the task solution type and (2) the concept correspondence type. In the task solution type, multiple agents are prepared, and the tasks are solved by communicating each other's situations and actions. The agents materialize words (instructions) to generate appropriate actions (Mordatch & Abbeel, 2018; Foerster et al., 2016). Additionally, the tasks are solved through combinations of agent communication by generating answers to specific questions, such as VQA methods (Das et al., 2017b; Evtimova et al., 2018; Sukhbaatar et al., 2016). The concept correspondence type generate messages corresponding to objects recognized through a reinforcement learning(RL)-based approach (Havrylov & Titov, 2017; Lazaridou et al., 2016; Cao et al., 2018; Lazaridou et al., 2018). This approach comprises different modules for generating and learning messages. It converges on a common language because the reward is transmitted to each module. However, the RL-based approach suffers from internal consistency (Choi et al., 2018; Kottur et al., 2017; Bouchacourt & Baroni, 2018; Lowe et al., 2019).

Choi et al. (2018) proposed the obverter technique to generate compositional languages. The obverter technique, which was proposed by Batali (1998), can solve "the problem of internal consistency in languages" in the RL-based approach and enables language emergence based on "the theory of mind (Premack & Woodruff, 1978)." However, in Batali's study, experiments were conducted under strong constraints that allowed the corresponding concepts to be uniquely determined. Language emergence under conditions without disentanglement information has not been studied. In Choi et al's study, supervised signal was required for generation.

Our study is inspired by the studies of Batali (1998) and Choi et al. (2018) . In this study, we extend the abovementioned studies to conceptually generate corresponding massages with formal rules without a human supervisor. The formation process and concept correspondence of human communication are useful as reference. Human language acquisition does not involve a supervisor in generating new words . Furthermore, it is not characterized by not learning concepts and words simultaneously (Imai & Masuda, 2013; Milligan et al., 2007). The emergence of sign languages in Africa (Senghas et al., 2004) and the formation of languages such as the Pidgin and Creole languages demonstrate the unsupervised and asynchronous nature in language systems. Additionally, it is known that animals acquire different formalities depending on the species (Spierings & ten Cate, 2016).

# 3 METHOD

## 3.1 MULTI-AGENTS DESCRIPTION GAME

The description game consists of a teacher agent that generates messages and a student agent that receives and learns the messages (Lazaridou et al., 2016; Choi et al., 2018). The student receives the messages from the teacher and associates them with the presented image. Repeating this process creates a language with concepts and message shared between agents.

We herein propose the multi-agent description game to establish a language system for multiple agents without a human supervisor. The game proceeds as follows for each epoch. (1) One teacher is randomly selected from all agents. (2) The teacher agent randomly selects one image from a list and generates the corresponding messages. (3) The teacher agent sends the image and message to the student agent, and the student agent associated the message with the image. (4) (1) to (3) are performed a specified number of times. One hundred agents are prepared.

No image labels are presented between in-game agents. Therefore, the agent must associate the message with the representation read from the image. There is no guarantee that a message will completely correspond to a label[1].

The image object used in this study was the MNIST database. A training dataset was used during learning and generation, and a test dataset was used when evaluating the messages.

## 3.2 MODEL ARCHITECTURE AND OBVERTER TECHNIQUE

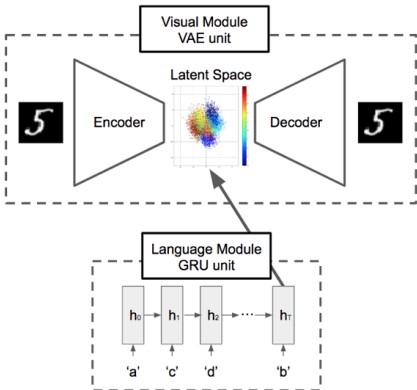

Figure 1: **Agent model architecture.** The visual module contracts the VAE unit. The language module contracts GRU units. Visual module is composed of an encoder and a decoder. The encoder output is mapped to the latent space. The language module generates and learns a message that corresponds to a point in the latent space.

We herein propose an agent architecture for the multi-agent description game. An agent comprises a language module for reading and generating a message, and a visual module for processing an image and associating the image with the messages. Each agent cannot exchange information other than messages and images. Figure 1 shows the configuration of the agent module.

The visual module uses a variational autoencoder (VAE) (Kingma & Welling, 2013) and the language module uses a gated recurrent unit (GRU) (Cho et al., 2014). The visual module has prior knowledge of the MNIST. The language module outputs a numerical value corresponding to the latent space by inputting an arbitrary message that has been converted into a one-hot vector. This structure enables the generation of messages corresponding to the image and reconstruction of a Dream (Ha & Schmidhuber, 2018) that corresponds to the messages. Dream is defined as an output image reconstructed by a message. It is not an output image reconstructed from the VAE input

---

[1]It is noteworthy that "image" means raw pixel data that are provided to the agent, and "label" means "numbers (common concepts understood by humans)" written to the image.

(the output image is expressed as Reconstructed Image). The details of the model architecture are described in Appendix A.

This module architecture differs from the preexisting architecture in terms of three characteristics: (1) The proposed architecture enables language emergence by agent interaction without using human-prepared supervised signals and rewards. (2) The concept corresponding to the generated message can be directly confirmed. (3) We can explore the structure and characteristics of messages organized by agent interaction.

An agent learns and generates a language using the obverter technique (Batali, 1998). In the obverter technique, at the time of generation, assuming that a teacher has a mind similar to that of a student, the teacher generates a message that maximizes his own understanding. In the teacher agent, the model parameters are fixed. When a target image is input, the visual module outputs a latent space value z corresponding to the image. After initializing the GRU hidden layer, the language module evaluates the output of all possible symbols. During this evaluation, the symbol with the minimum value between latent space z and the output of the language module is selected. The GRU hidden vector derived by the selected symbol is used by the next GRU unit. This procedure is repeated until the minimal value between z and the output value of the language module is less than a predefined threshold or until the max message length is attained. In the student agent, the GRU model parameters are updated. The parameters are updated by backpropagating the mean square error loss between the output of the GRU unit and the latent space z. These algorithms are described in the Appendix B.

The abovementioned procedure corresponds to self-supervised learning from the viewpoint of one agent. An agent set takes an emergent behavior that forms a conceptual pact without supervised signals.

## 3.3 INTERNAL REFLECTION FUNCTION

We introduced cognitive bias to the agents. In human conversations, not all words reach another human, and in some cases, they are rejected by composing them with one's own concept (Brennan & Clark, 1996; Lake et al., 2019) . This function means that the student compares the messages generated by the teacher. The student agent rejects the message if it is significantly different from the self-concept. In this study, we incorporated this function as the "internal reflection function" into the agents.

The student agent performs the learning so that the received message corresponds to the image. The internal reflection function algorithm was built into the student agent's learning. (1) The student agent receives an image that the teacher agent used to generate a message. Using the image, each student agent generates a message by the obverter technique shown in § 3.2. (2) Each student agent calculates a degree of comparison similarity between the message received from the teacher agent and the self-generated message using gestalt pattern matching (Ratcliff & Metzener, 1988). The degree of comparison similarity by gestalt pattern matching is output in the range of 0-1. The value of 1 means an exact match. (3-A) When the degree of comparison similarity is lower than the threshold value, the received message is regarded as an error and learning is not performed. (3-B) When the degree of comparison similarity is higher than the threshold value, the received message is regarded as correct and learning is performed.

In this experiment, three objects were prepared for comparison. The first object was **NORMAL**: internal reflection function. The second object was **INTERNAL**: an internal reflection function was incorporated from the beginning of learning. The threshold degree of comparison similarity was set to 0.4. We tested each learning object five times and evaluated the results.

## 3.4 EVALUATION METHODS

We analyzed the language generation characteristics of each learning method. In particular, we investigated the correct answer rate of Dream, which the agent reconstructed from the messages, and the systematic formal rules of the messages. These points are necessary for the compositional properties of the emergence language.

The first analysis was to confirm that learning has converged and to examine the similarity of messages between labels. We verified the mean square error loss between the output of the GRU unit and the latent space z. We compared the similarity of the same inter-label and different inter-label by Jaccard similarity coefficient. For each agent, 100 images were presented and messages were generated by obverter technique. The messages for each label are compared. We calculate the Jaccard similarity coefficient between labels for all agent combinations. Since there are 100 agents in an agent set, a 100 x 100 Jaccard similarity coefficient matrix is generated for each inter-label. The matrix represents the similarity between labels. Then, to derive the similarity between labels of the agent set, We calculate the average of each matrix. By dividing each the value into the same inter-label group and different inter-label group and taking their average values, we confirmed the similarity of messages between labels in each agent set. We tested three times using different images for each agent set.

The second analysis was the evaluation of Dream. We entered the message into an agent to evaluate the reconstructed Dream. The message was generated as follows. Three arbitrary agents were selected from each agent set, and 200 symbol strings were generated. For each agent, 100 messages were generated using a common image among the agents. Another 100 messages were generated using independent images for each agent. Validity was evaluated using five MNIST classifiers with over 98% accuracy, using Vedantam et al. (2018) as reference.

In the third analysis, the characteristics of lead symbol composition were compared with label and head symbol analyses. The analyses comprised 600 messages generated in the 2nd evaluation.[2]

In the label analysis, the composition ratio of the lead symbol to one label was confirmed. If one label ' s lead symbol consisted of only one or a few symbols, a clear owning relationship existed between the symbol and label. However, if it was composed of a plurality of lead symbols, no common rule based on the lead symbols existed. This analysis indicated the stability of the generated symbol. Specifically, the number of labels whose lead symbol constituted more than 70% of each label was counted.

In the head symbol analysis, whether the generated head symbol is biased to a specific symbol is evaluated as a variance. If the head symbol was generated without bias (i.e., using various symbols for the head), the variance was small. However, if a specific head symbol was commonly (monopoly) used (i.e., using only a single symbol), the variance was large. The variance formula is expressed as follows. $var = \frac{1}{n}\sum_i^n (x_i - \mu)^2$, where $x_i$ represents the number of uses of a certain head symbol and $n$ represents head symbol's number. The equality and division properties of the generated symbols were evaluated in this analysis.

Using these two indexes, the characteristics of the generated messages were evaluated quantitatively (Brighton & Kirby, 2006).

The fourth analysis confirmed the formal rule for the message corresponding to Dream. We entered a message to each agent set and confirmed the conditions to be reconstructed. Language structure by recursion and multiple symbols is the basic element of "Compositionality" in language emergent research (Kirby & Hurford, 2002). Therefore,We attempted to describe repeated patterns that appeared in the message as a formal rule.

The analyzed formal rules such as pattern and key symbol were evaluated from a human criterion. Therefore, we evaluated whether the evaluation of human formalization conformed to the agent's viewpoint. An index exists from the agent's viewpoint: Similarity evaluation between generated messages and formalized messages. We analyzed the degree to which the messages formalized from the human viewpoint were utilized among the agents. Three different messages based on the formal message were prepared as the criterion of similarity degree 1. Each agent set input an image and generated a message. The similarity between the generated and formal messages was compared using a histogram. A total of 10000 images were prepared.

Next, we simplified the formal rules and analyzed whether a specific syntax appeared in the generated rules. Simplification omits parentheses and repetitions. The formal rules were aligned for each leading symbol. The analysis was focused on the head symbol and the unique symbol. In the

---

[2]Hereinafter, when the first symbol in the message is indicated or when a general concept is used, it is referred to as "Lead symbol". The first two-digit symbol, which is frequently used in analysis, is referred to as "Head symbol".

Table 1: **Symbol identity among labels using Jaccard coefficient.** Same label means the average of the Jaccard coefficients between the same labels. Other label means the average of Jaccard coefficients between different labels. The s/o ratio is the proportion between the Same label and the Other label. No statistically significant difference was observed between the learning objects.

| OBJECT | Same label | Other label | S/O raito |
|---|---|---|---|
| NORMAL | 0.10058467 | 0.00776912 | 12.9832234 |
| INTERNAL | 0.095 | 0.01014133 | 10.5054888 |

unique symbol, symbol duplication was confirmed by comparing the formal rules of the same head symbol. In the head symbol, it was confirmed that the first two-digit symbol did not overlap with other formal rules. Using these indices, the structure of the formal rules in each learning object was confirmed.

## 4 RESULT

### 4.1 BASIC INFORMATION EVALUATION

We confirmed the characteristics of the learning objects in the message generation and learning phases. First, the distance between the message and target image was analyzed using the MSE. The loss tended to decrease during advanced learning in all learning objects (Figure 2). In INTERNAL, a gradual decrease was shown compared with NORMAL. NORMAL, which had no internal reflection function, showed an upward trend from epoch = 850. In INTERNAL, this tendency was not observed until epoch = 1100 but turned slightly upward at epoch = 1200.

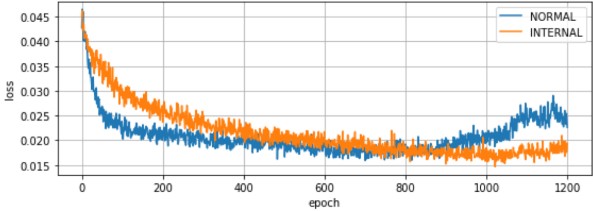

Figure 2: **Transition of teacher's loss function.** The average of five trials is presented. Averages from epoch = 700 to epoch = 900 were 0.018 and 0.017 for NORMAL and INTERNAL,respectively. Meanwhile, averages from epoch = 1100 to epoch = 1200 were 0.023 and 0.017 for NORMAL and INTERNAL, respectively. A clear increase in loss function was observed only in NORMAL.

Thus, these data exhibited different loss function transitions depending on the learning objects. Subsequently, we attempted to analyze epoch points that have a smaller loss function value for each learning objects. (NORMAL = 810, INTERNAL = 1200. For the analysis with NORMAL = 810, we created five new verification agent sets at epoch = 810. The decrease in loss function showed a similar trend.) Next, symbol identity by each label was analyzed using Jaccard coefficients. All learning objects exhibited tendencies toward symbol similarity for the same label and tendencies toward exclusiveness for different labels (Table 1).

### 4.2 CONVERT SYMBOL TO DREAM

We confirmed the accuracy of Dream reconstruction for the message. In all the learning objects, the average accuracy was 0.47- 0.50 (Figure 3). Statistically significant differences in the accuracy were not observed. The clustering accuracy of the VAE latent space was approximately 0.45 using the v-score. Therefore, Dream reconstruction by the message exhibited similar ability as that of latent space clustering. Shareability was confirmed as well.

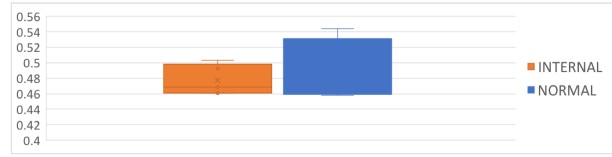

Figure 3: **Dream reconstruction accuracy in each learning object.**

Table 2: **The label analysis and the head symbol analysis.** Each value is obtained from the average of five trials.

|                      | NORMAL  | INTERNAL |
|----------------------|---------|----------|
| head symbol analysis | 1041.42 | 559.08   |
| label analysis       | 5.8     | 6        |

In additional, the accuracy of the labels by the learning objects was compared. In INTERNAL, the difference in the maximum and minimum values of the accuracy among the labels was small. However, NORMAL exhibited some degree of dispersion between the minimum and maximum values. An example messages whose label Dream is to be reconstructed is shown in Appendix E.

### 4.3 FORMAL EVALUATION OF LEADING SYMBOLS

From the label and head symbol analyses, characteristics of lead symbol constructability are compared. Table 2 (the label analysis) showed almost the same value at the end of learning in the learning objects. No statistically significant difference was observed. Meanwhile, in the head symbol analysis, each learning object showed different trends. At epoch = 210, the variance was almost equal in all learning objects. In NORMAL, the variance remained almost constant. INTERNAL showed a downward trend at each epoch. Finally, a statistically significant difference appeared in the value of the head symbol variance between NORMAL and an internal reflection function such as INTERNAL.The existence or nonexistence of the internal reflection function caused the lead symbol to exhibit a quantitatively different structure.

### 4.4 ATTEMPTS TO FORMALIZE MESSAGES

### 4.4.1 DECODING FROM HUMAN VIEWPOINT OF SYMBOL FORMAL RULE

One agent group was extracted from each learning object. We tested the formal rule of the message. The formal rules for each label are shown in Table 3.

In all learning objects, each agent set formed its own formal rule. A key symbol was created to generate each label’s Dream. In NORMAL, for example, the formal rule to be reconstructed with label “1” required the symbol “d” that is at least five digits long. In INTERNAL, labels “4,” “7,” and “9” began with the same symbol, and each label comprised a repeated pattern of different symbols. A formal rule having repeated patterns including the order and combination of symbols emerged in INTERNAL. Additionally, Internal identified the presence of exclusive symbols that prevented the label from being reconstructed. Meanwhile, NORMAL had few labels with such a formal rule. NORMAL showed few clear repeated pattern and could reconstruct Dream by the minimum necessary key symbol. A sample of Dream reconstructed by the message with formal rules is presented in the Appendix G. When the similarity is low, we confirmed in Appendix H that INTERNAL with a repeating pattern contributes effectively to the reconstruction of Dream.

Next, we confirm whether the formal rule message decoded from the human viewpoint is generated as the mainstream message in the agent set above. The results are shown in Table 4. In all learning objects, half of the labels had similarity peaks of 0.55 or higher. The results showed that the similarity between a message with a formal rule decoded by a human and that generated by an agent could be confirmed. We compared the characteristics of each label in the learning objects.

Table 3: **Comparison of formal rules.** We analyzed the message in which Dream reconstruction is guaranteed for each label. One agent group was extracted from each learning object. [xxx]* means that the xxx symbol is repeated. "y" symbol of [xxx]*y means an unrepeated symbol. ((z) means the symbol that may be required depending on the repetition state of the messages. "without s" means an exclusive symbol that cannot be reconstructed if the symbol exists.

| LABEL | NORMAL | INTERNAL |
|---|---|---|
| 0 | [aaa]* | [dda]*,[da]* |
| 1 | d>=5 | (a)[bba]*b |
| 2 | [dcab]*,[dacb]* | (db)[aaa]* |
| 3 | [dad]*,[dda]* | (aa)[bdb]*a, |
| 4 | [bbc]*a | [cad]*, [ca]*d, |
| 5 | (d)cccacca | d[bd]* |
| 6 | [bba]* | d[aaa]*d,a[aaa]*d |
| 7 | b>=6 | [cc(c)a]*,(without "d") |
| 8 | ccd | [bbcd]*,[bbd]* |
| 9 | [bcbbc]* | [ccb]*,[cbc]* |

Table 4: **Similarity evaluation between generated and formalized messages.** We constructed a histogram for each label and analyzed the peaks. Comparing the formal rule with the generated symbol, we discovered labels with one peak value and labels with bimodality. We considered that the higher the peak value, the more similar tendencies existed in the formal rule.

| | NORMAL | INTERNAL |
|---|---|---|
| Over 0.55 | 5 | 6 |
| Bimodal peak | 2 | 1 |

NORMAL tended to show a low similarity peak when of plural symbols are combined in the form rule. However, In INTERNAL ( except label "4" in INTERNAL that was at a low formal rule peak of 0.4.), labels using multiple symbols showed high similarity peaks.

### 4.4.2 PARSING OF FORMAL RULES

After simplifying the formal rule shown in Section 4.4.1, the syntax of the generated rule was analyzed (Table 5).

Table 5: **Formal rules aligned for each leading symbol.** To verify the structure, the formal rules were aligned for each leading symbol (left, NORMAL; right, INTERNAL). If the first symbol up to the second digit in the same series is unique, then that part is indicated by an underline. Symbols that are not used elsewhere in the same series are shown in italics.

| a series | b series | c series | d series | a series | b series | c series | d series |
|---|---|---|---|---|---|---|---|
| aaa | bbbbb | ccc*a* | dad/dda | aaa | bbab | ca*d* | daaad |
| | bba | cc*d* | da*cb*/ dcab | | bb*c*d/bbd | ccac | dbd |
| | bbca | | ddd | | bdba | cc*b*/ c*b*c | dda/da |
| | bcbbc | | | | | | |

**NORMAL**: the "b" series was frequently used in the lead symbol. Three of the four were composed of the same head symbol. In addition, no unique symbol existed for each formal rule. The head symbol was the same in the "c" series. However, a unique symbol existed for each formal rule. In the "d" series, the unique symbol and the head symbol appeared on the same formal rule.

**INTERNAL**: The lead symbol is allocated almost evenly. In the "b" series, a distinctive head symbol appeared, while the unique symbol appeared in one formal rule. In the "c" series, a distinctive head symbol appeared in two formal rules, while the unique symbol appeared in two rules. In the "d" series, a distinctive head symbol appeared in two rules, while the unique symbol appeared in one rule.

In INTERNAL, head symbol duplication was less than in NORMAL. The number of unique symbols appeared was almost the same in each learning object. However, in the "c" series of NORMAL, the unique symbol appeared as parenthesized symbols in the formal rules of each other and are not completely independent symbols. Even in the "b" series, which requires a clear divisions, only one formal rule appeared clearly independent. Meanwhile, in INTERNAL, distinctive head symbols and unique symbols appeared in a balanced manner in each rule. Each rule had a proprietary tendency. It was confirmed that even a formal rule not having a unique symbol exhibited a different pattern.

Therefore, the tendency to form a unique symbol combination in INTERNAL was confirmed. The spatial representation of the message is shown in Appendix I. It represented the representational difference for each message in the agent.

## 5 DISCUSSION

In this study, we proposed a multi-agent description game with 100 agents and a model architecture. The obverter technique was used for message generation. In addition, the internal reflection function, which imitates human language acquisition, was introduced.

As described in Sections 4.1 and 4.2, messages corresponding to the label was successfully generated, and the reconstruction of Dream was confirmed from the messages. It was found that by the proposed method, messages that correspond to concepts and allow an agreement between agents emerged without any supervisor. However, as reported in Section 4.1, the decreasing tendency of the loss function was different between learning objects. As described in Sections 4.3 and 4.4, the formal rule between learning objects differed. In NORMAL, a rule responding to a specific symbol element is generated, while in INTERNAL, a symbol combination pattern is generated. Each label has its own combination pattern. It was found that the formal rules of messages that could be conceptually agreed by internal reflections were structured differently. This result suggests the existence of qualitatively different formal trends caused by internal reflection function.

Therefore, we considered the features of the internal reflection function. NORMAL allowed students to learn all the messages generated by the teacher as a "positive example." In this situation, the "this and that" symbol was learned; therefore, even a slight difference was absorbed as a "correct" message. The results of the loss function indicated that this situation occurred in "overlearning."

Meanwhile, in the internal reflection function, a learning constraint based on the agent's similarity evaluation existed. Because of this constraint, it is presumed that each agent creates a "negative example" in a pseudo manner by comparing itself with others. The positive and negative examples might have constructed a pseudo boundary among the labels. Therefore, we assumed that a conservative message has been constructed. Consequently, a whole agent tends to a generate message having a common symbol for the same label. This situation, in which only common-term messages are generated, could be regarded as a bottleneck in information transmission between agent groups. Vogt (2005) reported that bottlenecks (Kirby, 2001; Brighton, 2002; Smith et al., 2003) produced syntactic structures. The bottleneck was considered to have resulted in the generation of massages having a pattern in the formal rule.

An additional experiment was conducted on "SWITCH" in Appendix C. For SWITCH, the learning start point was NORMAL, and after some learning steps, the internal reflection function was activated. The results are shown in the Appendix. SWITCH showed the same results as INTERNAL. SWITCH was assumed to have broken the language structure at the time of NORMAL; thus, a new formal rule emerged. Therefore, the internal reflection function can be expressed as an emergent parameter of a conceptually agreed language system.

In this study, we established a method for the emergence of a language system, in which consensus could be achieved in a group without a human supervisor. We proposed an emergence parameter, i.e., the "internal reflection function," which acquired structuring formal rules; additionally, we con-

firmed its effectiveness. This suggests that it performs effectively under-restricted situations similar to human language generation. Further studies are required to generalize the relationship between formalization and concept. We confirmed the laws of systematization generated by machines using data other than those of MNIST and having "different concepts" in each agent.

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

## A  MODEL ARCHITECTURE

We used PyTorch for all of our implementations. The module is structured independently. This was constructed with reference to conceptual correspondence research (Higgins et al., 2018; Vedantam et al., 2018) and translation method by Pivoting (Gella et al., 2017; Lee et al., 2018). Among them, SCAN (Vedantam et al., 2018) connected the concept of latent space disentangled by bata-VAE (Higgins et al., 2017) with the corresponding words using the KL divergence. Consequently, image areas corresponding to human words (nouns, adjectives, and conjunctions) can be automatically estimated from raw pixel data without annotation. Furthermore, SCAN has successfully reconstructed images that were not directly learned from supervisor data. To create a language system without a supervisor, which is the purpose of this study, methods using concepts and symbols will be useful as reference.

### A.1  VISUAL MODULE

We used VAE (Kingma & Welling, 2013) to learn MNIST. (1) Encoder side. The input layer is composed of 782 dimensions. The activation function used the ReLU. The next hidden layer consists of 512 dimensions. To represent the latent space, the sampling layer that generates the latent space z is connected to the hidden layer. In this study, the latent space is set to four dimensions. (2) Decoder side. The decoder consists of a hidden layer that converts four-dimensional information in the latent space to 512 dimensions and an output layer of 782 dimensions. The activation function of the output layer is a sigmoid function. The optimization function is Adam (Lr = 1e-3). For creating the agent, 100 visual modules were created by learning different initial values.

### A.2  LANGUAGE MODULE

We used a single layer GRU (Cho et al., 2014) to implement the language module. The size of the hidden layer was 128. The GRU uses the many-to-one output method. The GRU outputs a four-dimensional value to any input symbol string. The optimization method was SGD (Lr = 0.005). The initial value of the language module was different.

# B    MESSAGE GENERATION BY MULTI-AGENT DESCRIPTION GAME AND OBVERTER TECHNIQUE

The obverter technique algorithm at the time of generation is as follows algorithm 1. We improved the original paper (Batali, 1998).

---

**Algorithm 1:** Message generation process used in our study

---

$\mathbf{h}^{(0)} = 0$ //Initialize GRU hidden layer with zeros;
s = [ ] //Initialize the message vector;
t = 0 //Timestep index;
$\mathbf{V} = \mathbf{I} \in \mathbb{R}^{(4 \times 4)}$ //Each row $v_0$, $v_1$, $v_2$, $v_3$ corresponds to a, b, c, d;
$\mathbf{z}$ = VisualModule ' s latent space($\mathbf{x}$);
**while** $\mid s \mid <$ *max message length* **do**
  $\mathbf{h}_{v_i}^{(t)}$ = Language Module($v_i$,$\mathbf{h}^{(t-1)}$);
  i = $argmin_i$ MSE($\mathbf{z}$, $\mathbf{h}_{v_i}^{(t)}$);
  $\mathbf{h}^{(t)} = \mathbf{h}_{v_i}^{(t)}$;
  Append i to s;
  **if** *MSE ($\mathbf{z}$,$\mathbf{h}^{(t)}$) < threshold* **then**
    Terminate;
  **end**
**end**

---

Control parameters consisted of the max message length and a threshold value. In this experiment, the max message length was 10 and the similarity threshold was 0.005. In learning, the role of a teacher is randomly selected for each epoch. One hundred images were presented for each epoch, and the teacher randomly selected and generated a corresponding message. Students learned using the message. When it reached the defined number of times, the next epoch was tested. In this experiment, the defined number of times was 100.

# C    VALIDATION OF SWITCH

To verify the strength of the constraints of the internal reflection function, we activated the internal reflection function during learning. This object was called **SWITCH**.

## C.1    METHOD

SWITCH required parameters for the similarity threshold and a starting point for the internal reflection function. The parameter used is a threshold value of 0.6/start at epoch = 600. Five agent sets were created.

## C.2    BASIC INFORMATION EVALUATION

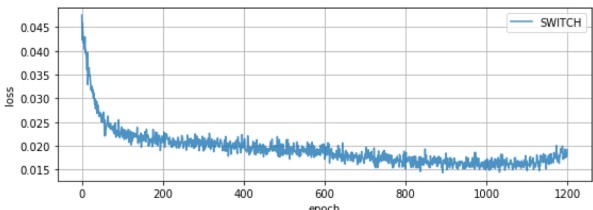

Figure 4: **Transition of teacher's loss function.** The average of five trials is presented. Average from epoch = 700 to epoch = 900 and from epoch = 1000 to epoch = 1200 is 0.016.

First, the distance between the message and the target image was analyzed by the MSE (Figure 4). SWITCH and NORMAL showed the same decreasing trend because they had no internal reflection function at the beginning of learning. In SWITCH, the loss function did not increase as in NORMAL after epoch = 800, but turned slightly upward at epoch = 1200. Owing to the internal reflection function, the trend differed from that of NORMAL. The analysis was performed at epoch = 1200.

Table 6: **Symbol identity among label using Jaccard coefficient.**

| OBJECT | Same label | Other label | S/O raito |
|--------|-----------|-------------|-----------|
| SWITCH | 0.08982 | 0.009324 | 11.0660784 |

In Jaccard analysis, the trend of SWITCH is the same as those of other learning methods (Table 6).

## C.3 CONVERT SYMBOL TO DREAM

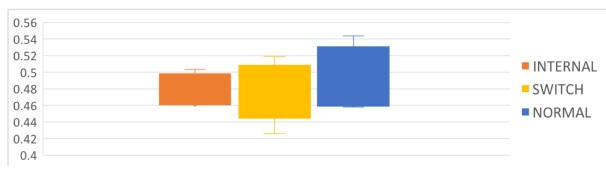

Figure 5: **Dream reconstruction accuracy in each learning object.**

We confirmed the accuracy of Dream reconstruction for the message. In SWITCH, the average accuracy was 0.47–0.50 (Figure 5). No statistically significant differences were observed in the accuracy. SWITCH tended to exhibit some degree of dispersion between the minimum and maximum values.

## C.4 FORMAL EVALUATION OF LEADING SYMBOLS

Table 7: **The label analysis and the head symbol analysis.**

| | SWITCH |
|---|---|
| head symbol analysis | 464.9 |
| label analysis | 4.8 |

The lead symbol in Table7 was analyzed. In label analysis, NORMAL and INTERNAL showed similar results. No statistically significant difference was observed. Meanwhile, the head symbol analysis showed a characteristic trend. SWITCH showed the same value as NORMAL before the internal reflection function was applied (before epoch = 600). However, after the internal reflection function was applied, SWITCH showed a similar value to INTERNAL. A significant difference was observed with NORMAL but not with INTERNAL.

## C.5 ATTEMPTS TO FORMALIZE MESSAGES

### C.5.1 DECODING FROM HUMAN VIEWPOINT OF SYMBOL FORMAL RULE

One agent group was extracted from SEITCH agent set. The formal rules for each label are shown in the Table 8. In SWITCH, an agent set forms its own rule system. Labels "0", "5", and "6" began with same symbol, and each label comprised a repeated pattern of different symbols. The characteristics of the formal rule were similar to Internal.

Next, we confirm whether the formal rule message decoded from the human viewpoint is generated as the mainstream message in the agent set above (Table 9). As in INTERNAL and NORMAL,

Table 8: **Comparison of formal rules.**

| LABEL | SWITCH |
|---|---|
| 0 | [a]* |
| 1 | [bd]*,[db]*,(without"a") |
| 2 | bbd(a)(b)b,(without"c") |
| 3 | d[ad]*,[dda]* |
| 4 | [cca]* |
| 5 | [aadd]*,[aad]* |
| 6 | [acb]* |
| 7 | [cb]* |
| 8 | [dda]*(d)c |
| 9 | [ccd]*[cd]* |

SWITCH tended to exhibit a high peak value. The prototype messages of SWITCH were composed of a combination pattern of multiple symbols. SWITCH was particularly prone to exhibit bimodal peaks.

Table 9: **Similarity evaluation between generated and formalized messages.**

| | SWITCH |
|---|---|
| Over 0.55 | 4 |
| Bimodal peak | 3 |

### C.5.2 PARSING OF FORMAL RULES

Table 10: **Formal rules aligned for each leading symbol.**

| a series | b series | c series | d series |
|---|---|---|---|
| aaa | bd | cca | dad/dda |
| aad/aadd | bbdb | cb | ddac |
| acb | | ccd/ cd | dda/ db |

After simplifying the formal rule shown in Section C.5.1, the syntax of the generated rule was analyzed (Table 10). SWITCH has a symbol syntax trend similar to that of INTERNAL. In "a" series, unique symbols appeared in two formal rules. A distinctive head symbol appeared in one form rule. In the "b" series, although the symbols were similar, different head symbols appeared. In the "c" series, a unique symbol existed for every rule. In the "d" series, a distinctive head symbol appeared in two rules.

## D   AGENT LEARNING PARTICIPATION TENDENCY BY INTERNAL REFLECTION FUNCTION

Figure 6 shows the agent learning participation tendency by the internal reflection function. With INTERNAL, the number of agents participating in learning increases with the epoch number. It has shifted to a plateau by epoch = 400. At the beginning of learning, the entire agent does not contribute to emergence. The language system understood by the entire agent was generated as learning progressed step by step. In SWITCH, the learning intermediate point was set as the internal reflection function start time. Because messages that can be reconstructed as Dream have been generated, messages corresponding to labels can be generated even with 70% learning participation.

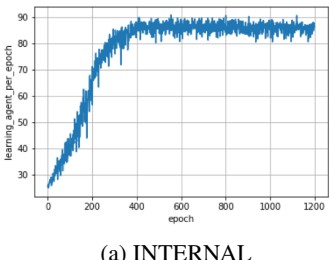

(a) INTERNAL

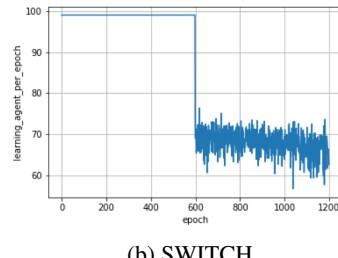

(b) SWITCH

Figure 6: **Agent learning participation tendency by internal reflection function**: Those figures show the average of five trials. One agent has the teacher role. Therefore, the maximum value for this graph is 99.

# E    EXAMPLE OF DREAM RECONSTRUCTED FROM MESSAGES

The quantitative evaluation in Section 4.2 simultaneously evaluates the accuracy and agent sharing for the label. We confirmed the subjective characteristics in the reconstructed Dream. Using an agent set that confirmed the formal rule of Section 4.4, we demonstrated that Dream corresponded to the input messages. Figure 7 shows the results of reconstructing Dream for all agents.

As shown, all learning objects have the same ability to reconstruct Dream. Dream was reconstructed on many labels. Some labels have low agent sharing for the messages. For example, labels "6," "8" for NORMAL, label "6" for INTERNAL, and labels "2," "6" for SWITCH showed this tendency.

However, we confirmed that all the same messages reconstructed Dream corresponding to the label.

# F    LEAD SYMBOL COMPOSITION ANALYSIS

Lead symbol composition is a reference for creating a formal rule. Therefore, referring to the symbol correspondence to the concept of Brighton & Kirby (2006), we confirmed the correspondence between the lead symbol and the label. Figure 8 shows that the symbol sequence was randomly generated regardless of the label at epoch = 0. Unlike that reported in the main text, the lead symbol compares only the first symbol. One agent set was extracted from each learning object. The composition state of the lead symbol and the symbol composition ratio with respect to the label are shown.

**NORMAL**: Figure 9 (Top) shows characteristic results. In label analysis, most labels have their own symbols. However, it was found that the owning symbol originated from a specific head symbol and was biased. This indicates that various labels concentrate on the same head symbol. This agent set shows that labels are concentrated on symbols "b" and "d."

**INTERNAL**: Figure 9 (middle) shows that the symbols are used equally. In this agent set, "c" corresponds to "4," "7," "9" and "d" corresponds to "0," "5," "6." As symbols are used more equally, in the label analysis, many labels established owning relationships with particular symbols.

**SWITCH**: Figure 9 (bottom) shows that the symbols are used equally as in INTERNAL. Additionally, owning relationships are shown from label analysis. However, it appeared that some labels were constructed with exactly half of the owning symbols (for example, labels "3" and "5" simultaneously used "a" and "d", and label "7" simultaneously used "b" and "c"). Labels with bimodality tended to appear in SWITCH.

# G    DREAM RECONSTRUCTION SAMPLE OF MESSAGES WITH FORMAL RULES

We confirmed the Dream that is reconstructed from the message, and established the formal rule. In each learning object, the formal rule was constructed based on the generated messages. We entered the message individually and built a formal rule based on the reconstructed Dream's relationship.

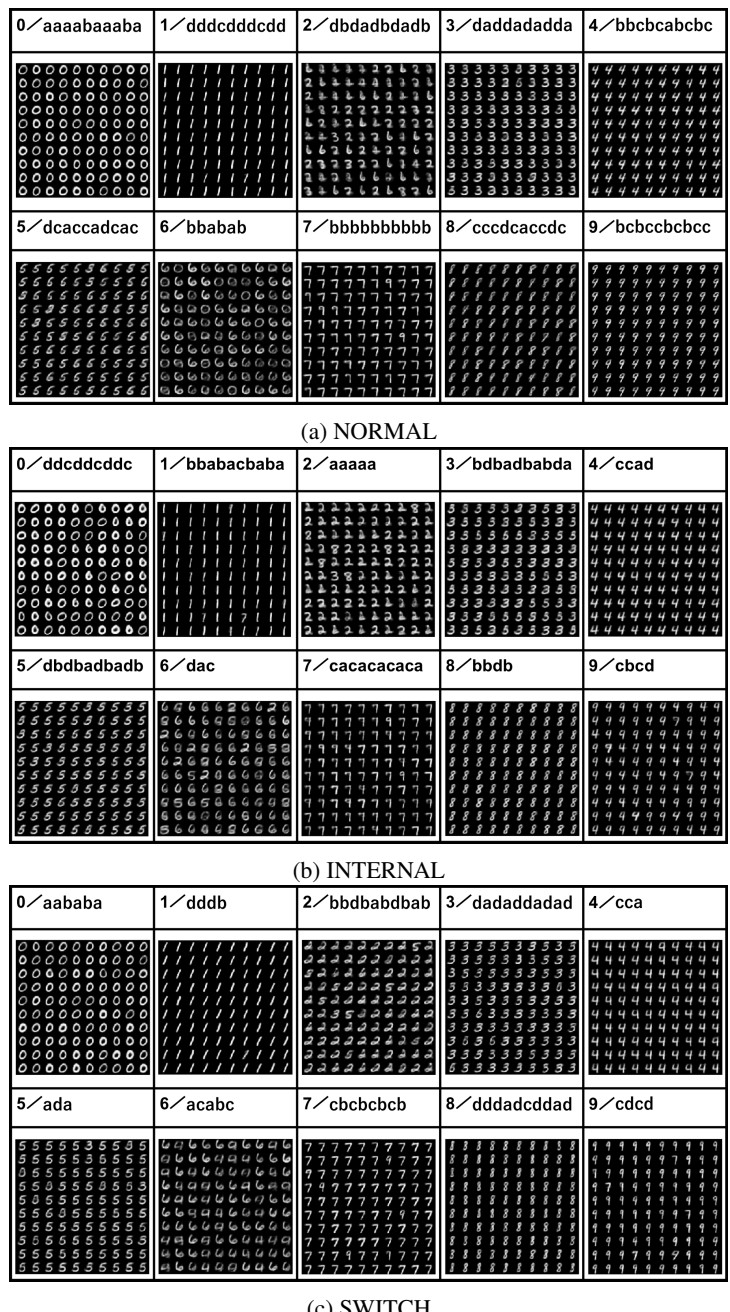

(a) NORMAL

(b) INTERNAL

(c) SWITCH

Figure 7: **An example of Dream reconstructed from messages**: The message below each figure was input into the agent. The numbers in each figure represent the corresponding label numbers.

Figure 10 shows a message for each label and the corresponding Dream. For each learning object, two characteristic formal rules are shown.

**NORMAL**: labels "0" and "7" are targeted (Figure 10, (a)). For label 0: Dream was reconstructed by message [a]*. However, label 0 could not be reconstructed when the number of symbols in "a" was small. A single "a" symbol was reconstructed for label "5." For label "7": Dream was reconstructed by message [b]*. However, if the number of "b" symbols was less than five digits, it was reconstructed to another label ("4" or "9"). Hence, NORMAL has a formal rule by the key

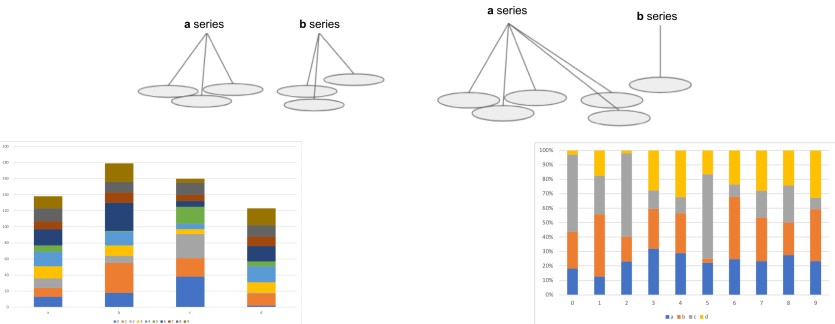

Figure 8: (Top) **Correspondence between concepts and symbols.** It is important that the symbols corresponding to the concept are not random, have no bias, and have a one-to-one correspondence. In this study, the correspondence is analyzed with the lead symbol. The case where one symbol excessively corresponds to the label or the case where a certain label forms the lead with a plurality of symbols is shown on the right. On the contrary, the situation where one symbol is appropriately classified and the case where the label is composed of one symbol at the lead are shown on the left. In this study, as 10 concepts (labels) must be represented by four symbols, 2.5 labels per symbol is ideal for a simple calculation. (bottom) **Correspondence between label and lead symbol composition at epoch = 0 in one agent set.** The figure on the left shows the configuration of the lead symbol. The figure on the right shows the configuration of the label analysis.

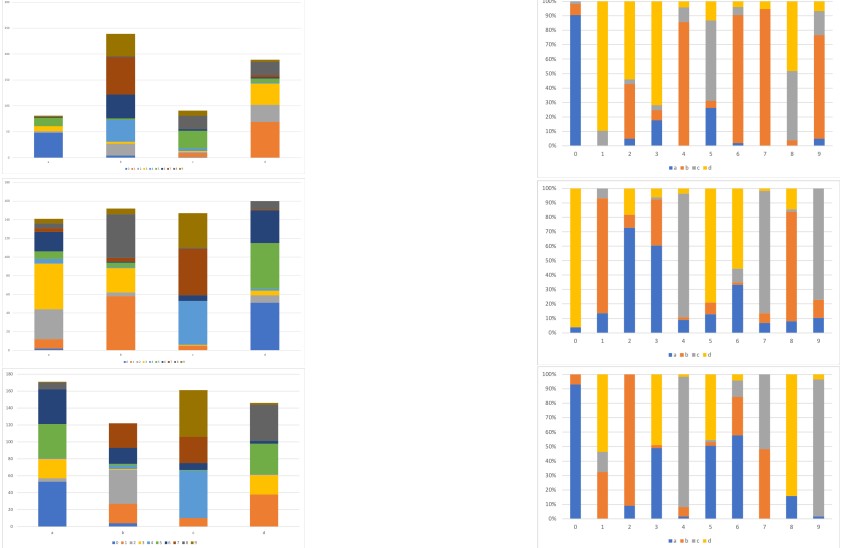

Figure 9: lead symbol composition analysis. We used the agent set in Section 4.4: NORMAL (top); INTERANL (middle); SWITHC (bottom). The figure on the left shows the lead symbol composition analysis. The figure on the right shows the label analysis.

symbol. This occurs when the reconstructed Dream is different owing to on the length of the symbol string. The formal rule includes the length of the symbol string.

**INTERNAL**: labels "5" and "8" are targeted (Figure 10, (b)). For label "5": "d + [bd]*" ' s formal rule appeared. A clear pattern was structured on this label; e.g., "dbddbd" could not be reconstructed. For label "8": "b," "c" and "d" are required in the formal rule. Since Dream was generated successfully, it was shown that it was patterned including the symbol "c". Hence, strict and robust labels existed for the pattern of the message to be reconstructed in Dream.

**SWITCH**: labels "2" and "9" are targeted (Figure 10, (c)). For label "2": The formal rule was most difficult to obtain. However, two symbol strings, such as "bbdba" and "bdbab," were successfully

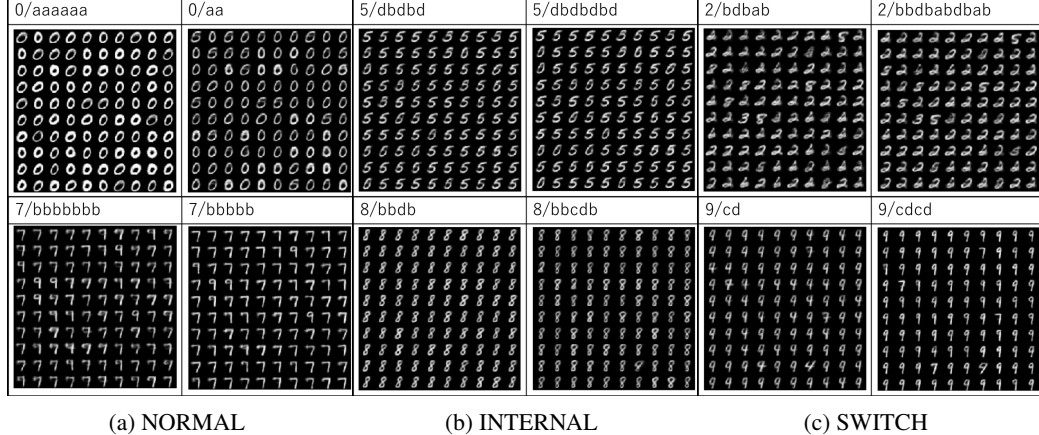

|  |  |
|---|---|
| (a) NORMAL | (b) INTERNAL |

(c) SWITCH

Figure 10: **Dream reconstruction sample of messages with formal rules.** We used the agent set in Section 4.4.

reconstructed in Dream. This was a characteristic formal rule of a five-digit symbol string as the key. For label "9": Dream was reconstructed for the message of [cd]*. The "c" symbol was the reconstructed Dream of labels "4" and "7." However, the "d" symbol showed a direction to label "9." For label "9," we confirmed that the combination "c + d" was the key pattern.

## H  CHARACTERISTIC OF MASSAGES WHERE DREAM RECONSTRUCTION IS SHOWN AT THE SAME SIMILARITY

The condition of the message in Dream reconstruction was confirmed from messages with similarity 0.5. Because a message with low similarity was distant from the messages to be reconstructed, the Dream reconstruction was considered difficult. Hence, we analyzed the pattern of the message to be reconstructed as Dream (Figure 11).

In all the learning objects, even if the degree of similarity of the formal rule was approximately 0.5, messages capable of Dream reconstruction existed. Dream reconstruction was observed in messages comprising patterns or symbols according to the formal rule. The Normal and Internal reflection functions showed every different feature in the messages when the similarity was low.

In the Internal reflection function, Dream tended to be reconstructed when a repeated pattern of formal rules in the target message existed (Figure 12 showed labels "4" and "7" for INTERNAL, and labels "5" and "8" for SWITCH.) Meanwhile, in NORMAL, Dream reconstruction might not succeed even if formal rules exist repeatedly. A tendency to be reconstructed existed when the minimum necessary symbol appeared. In the Internal reflection function, patterns in the formal rules were inviting for the rebuilding of DREAM even when the degree of similarity was low. Meanwhile, in NORMAL, the symbols presented in the formal rule were the minimum requirements for Dream reconstruction.

## I  VISUALIZATION OF LANGUAGE MODULE REPRESENTATION

We confirmed the cluster structuring of the language module using the generated messages. Using t-sne (Maaten & Hinton, 2008), the four-dimensional output was compressed to a two-dimensional output (Figure 12). For evaluation, Agent No.10 was used. Messages with a similarity of 0.7 or higher were plotted. The parameters used in t-SNE were iter = 2500 and perplexity = 100.

When symbols in the message have different tendencies in each learning object, they are represented in a space separated for each label. However, when symbols in the message are similar, the tendency is different depending on the learning objects. INTERNAL and SWITCH showed signs of spatial separation even in messages with the same tendency. For INTERNAL, in the formal rule of label 0-5-6, "d" is a common lead symbol. However, a cluster was confirmed for each label. In the SWITCH

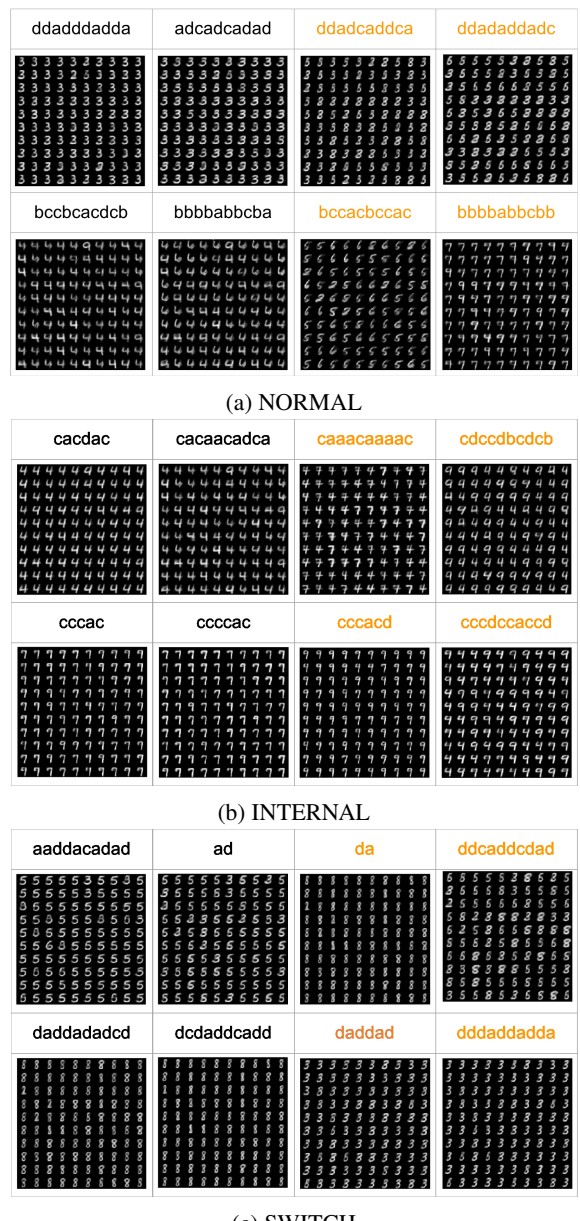

Figure 11: **Characteristic symbol sequence where Dream reconstruction is shown at the same similarity**:The number upper each figure is the corresponding label number. These Dreams were generated by the message above the figure. The figure on the left half (black) shows high accuracy. The figure on the right half (amber) shows the low accuracy.

4-7-9 formal rule, "c" is the lead symbol. Despite the overlap, clusters were created for each label. Similar trends were shown in INTERNAL label's 1-3-8 and SWITCH label's 1-2-7. The formal rule of those labels is constructed to repeat specific symbols.

Meanwhile, in NORMAL, a specific label could not be independent although a cluster was generated in a formal rule containing the same symbol. For example, labels "1" and "2" have formal rules sandwiched between symbols "d." However, the cluster at label "2" was included on top of label "1." Additionally, clusters were formed at the same positions as labels "5" and "8."

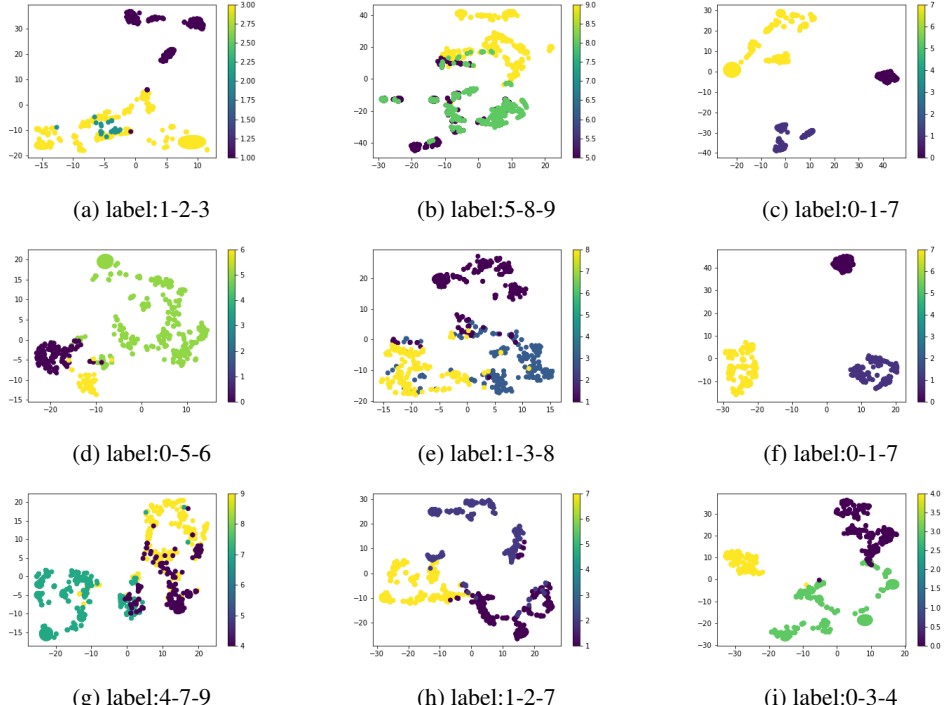

(a) label:1-2-3

(b) label:5-8-9

(c) label:0-1-7

(d) label:0-5-6

(e) label:1-3-8

(f) label:0-1-7

(g) label:4-7-9

(h) label:1-2-7

(i) label:0-3-4

Figure 12: **Visualization of language module representation.** We used the agent set and agent number in Section 4.4: NORMAL (a,b,c); INTERNAL (d,e,f); SWITCH (g,h,i). The number below each figure is the corresponding label number. The color depends on the corresponding label.

The internal reflection function shows that the generated messages not only appears as a formal rule, but also affects the representation space. We considered a new research topic for creating recursive relationships that disentangled the acquired representations according to the acquired language.

