# OpenReview forum: "CONTRIBUTION OF INTERNAL REFLECTION IN LANGUAGE EMERGENCE WITH AN UNDER-RESTRICTED SITUATION"
_ICLR.cc/2020/Conference — Reject_

### Official Review · AnonReviewer1 · 2019-10-22
**Official Blind Review #1**

**Rating:** 3

**Review:**

This paper proposes a framework for emergent communication system using internal reflection, similar to obverter technique in Batali 1998 and Choi et al., 2018, while not requiring explicit supervision. Despite the nice idea, the dire quality of writing makes it extremely difficult if not impossible to understand the proposed model and the results. Also, some of the experimental results seem unnecessary. In Table 3., for example, (1) in §4.4.1 the authors merely list the rules without giving an overall conclusion. (2) I'm not certain what's the significant of the results in Table 3., as these are messages from one trained instance, and I'm sure a different training run will yield different messages.

Although I am strongly leaning towards reject, I am willing to discuss with my co-reviewers if they have a different opinion.

Cons
- There doesn't seem to be a proper baseline in any of the experiments conducted.
- The writing generally is unclear, full of ungrammatical sentences and extremely hard to read: e.g. see examples below.
     - §3.2: "A language system can emerge by a pure machine, excluding the human viewpoint."
     - §3.2: "The structure, characteristics, and limitations of the messages generated by the machine can be determined."
     - §3.2: "The agents as a whole have a movement to create a system in which languages is unsupervised and one conceptual pact is formed."
     - §3.3: "Receive an image and generate a message based on self-knowledge."
     - §3.4: "This formal rule was from the human perspective".

- The evaluation section is very rambling and contains unnecessary parts that need not be included, e.g. convergence of mean squared error loss between GRU output and latent space z.

Questions
- The internal reflection function part, the core of this paper, is not properly explained until the end. For example, in §3.3, "when the comparison result is lower than the threshold value": what is the "comparison result"? How does one compare a message and another message? Please elaborate on this further.
- In the evaluation section §3.4, the authors keep referring to "dream": "correct answer rate of Dream", "We entered the message into an agent to evaluate the reconstructed Dream". But the authors don't provide a formal definition of "dream" except providing a citation to Ha and Schmidhuber, 2018. Are the readers expected to know what this means?
- In §3.4, "Jaccard coefficient were applied to test the message similarity between labels". How does one apply Jaccard index to compute similarity between labels? This seems an important part of evaluation but not explained at all.


**Experience Assessment:**

I have published in this field for several years.

**Review Assessment: Checking Correctness Of Derivations And Theory:**

I assessed the sensibility of the derivations and theory.

**Review Assessment: Checking Correctness Of Experiments:**

I assessed the sensibility of the experiments.

**Review Assessment: Thoroughness In Paper Reading:**

I read the paper at least twice and used my best judgement in assessing the paper.

---

> ### Author Response · Authors · 2019-11-13
> **Authors' response to the reviewers' comments.(part 1)**
>
> We thank you for your detailed review and helpful comments. We address your concerns as follows.
>
> - There doesn't seem to be a proper baseline in any of the experiments conducted.
>
> In current language emergence research, our sense is that there is no baseline as a common indicator because the model structure depends on the purpose. Therefore, we collected methods to find out the characteristics of the languages used in related research. We selected the ones that fitted the contents of this research.
> While, to confirm the effectiveness of the internal reflection function, a comparison was made based on the evaluation axis for the presence or absence of internal reflection. Like this, to evaluate the validity of the architecture, there is a study in language emergence research that evaluates architecture by controlling and disabling functions[1].
>
> - The writing generally is unclear, full of ungrammatical sentences and extremely hard to read.
>
> We have incorporated your comments, so we have clarified you pointed out.
>
>      - §3.2: "A language system can emerge by a pure machine, excluding the human viewpoint."(p. 4, lines 4-5)
> The proposed architecture enables language emergence by agent interaction without using human-prepared supervised signals and rewards.
>
>      - §3.2: "The structure, characteristics, and limitations of the messages generated by the machine can be determined."(p. 4, lines 6-7)
> We can explore the structure and characteristics of messages organized by agent interaction.
>
>      - §3.2: "The agents as a whole have a movement to create a system in which languages is unsupervised and one conceptual pact is formed."(p. 4, lines 22-23)
> An agent set takes an emergent behavior that forms a conceptual pact without supervised signals.
>
>      - §3.3: "Receive an image and generate a message based on self-knowledge."
> Please see our 4th explanation.
>
>      - §3.4: "This formal rule was from the human perspective".(p. 5, lines 41)
> The analyzed formal rules such as pattern and key symbol were evaluated from a human criterion.
>
> - The evaluation section is very rambling and contains unnecessary parts that need not be included.
> We appreciate the reviewer's concerns on this point. However, we consider our original text correct. We surveyed the evaluation method in related researches and selected indicators that can analyze the qualitative changes in language emergence by internal reflection. We consider that the features of the message, grammatical analysis, and the accuracy of the image reconstructed by messages were basic indicators for confirming the emergent language features in related researches. Thus, we would like to retain the original text.
>
>  - e.g.1: Analyze formal rules (§4.4.1 and Table 3)
> We obtained one trained instance from each learning object and analyzed formal rules of messages. By analyzing the structural differences in the formal rules for each instance, we are able to know the necessary symbol conditions and symbol patterns to generate a concept. This aspect is a necessary condition for any language to be considered compositional. On the other hand, We agree that different training will produce different messages. However, in other language emergence research, in addition to the quantitative characteristics of the message, they analyzed the language system in the instance to evaluate specific differences and compositional nature[2-4]. For this reason, we believe that to describe the generated language system would be more appropriate.
>
>
>  - e.g.2: Convergence of mean squared error loss between GRU output and latent space z (§4.1)
> Our sense is that just confirming the generated messages is not enough to know the relationship between concepts and messages. If messages have formal rules but are not grounded in the concept, the language is considered meaningless. Therefore, we need an evaluation index that confirms the correspondence between messages and concepts. MSE Loss is an index that evaluates the validity of the ground relationship between the generated message and the corresponding label.
>
> [1] Das, A., Kottur, S., Moura, J. M., Lee, S., & Batra, D. (2017). Learning cooperative visual dialog agents with deep reinforcement learning. In Proceedings of the IEEE International Conference on Computer Vision (pp. 2951-2960).

---

> ### Author Response · Authors · 2019-11-13
> **Authors' response to the reviewers' comments.(part 2)**
>
> Question
> - The internal reflection function
>
> We agree that this point requires clarification, and have added the specific algorithm to the method (p. 4,§3.3). “comparison result” is an ambiguous word. We changed the term to “degree of comparison similarity”. The degree of comparison similarity shows the similarity between the messages using “gestalt pattern matching”. We have also added the reference: John W. Ratcliff and David Metzener, Pattern Matching: The Gestalt Approach, Dr. Dobb’s Journal, page 46, July 1988.
>
>
> - A formal definition of "dream”.
>
> The reviewer's comment is correct. To clarify, we have added the following text to the MODEL ARCHITECTURE and OBVERTER TECHNIQUE(p. 3,§3.2, lines 32-33): Dream is defined as an output image reconstructed by a message. It is not an output image reconstructed from the VAE input (the output image is expressed as Reconstructed Image).
>
> - A Jaccard similarity coefficient
>
> The reviewer's comment is correct. To clarify, we have added the text to the evaluate method (p. 5,§3.4, line 1-12 ).
>
> [2]Choi, E., Lazaridou, A., & de Freitas, N. Compositional obverter communication learning from raw visual input. ICLR 2018
> [3]Havrylov, S., & Titov, I. (2017). Emergence of language with multi-agent games: Learning to communicate with sequences of symbols. In Advances in neural information processing systems (pp. 2149-2159).
> [4]Lazaridou, A., Peysakhovich, A., & Baroni, M.  Multi-agent cooperation and the emergence of (natural) language. ICLR 2017

---

### Official Review · AnonReviewer4 · 2019-11-02
**Official Blind Review #4**

**Rating:** 3

**Review:**

This seems to be an ambitious paper that claims to replicate the phenomenon of “spontaneous” learning of compositional language (as in Choi 2018) under relaxed constraints.  It extends the two-agent description game, which agents swap between Teacher and Student roles (“obverter technique”) into a multi-agent game in which messages can be rejected.

There are problems with clarity, particularly after page 4 (Sections 3.2 and 3.3), starting with the paragraph about how their architecture differs from the preexisting architectures. The lack of line numbers in the submission makes it impractical to give detailed feedback.

Overall I have had a lot of difficulty understanding the proposed method, and would have needed more time (or more background knowledge) in order to properly evaluate this paper.

**Experience Assessment:**

I do not know much about this area.

**Review Assessment: Checking Correctness Of Derivations And Theory:**

I did not assess the derivations or theory.

**Review Assessment: Checking Correctness Of Experiments:**

I did not assess the experiments.

**Review Assessment: Thoroughness In Paper Reading:**

I made a quick assessment of this paper.

---

> ### Author Response · Authors · 2019-11-13
> **Authors' response to the reviewers' comments.**
>
> We thank you for your detailed review and helpful comments. We address your concerns as follows.
>
> -There are problems with clarity, particularly after page 4 (Sections 3.2 and 3.3)
>
> Thank you for your comment. Please see our 2nd and 4th explanations for Reviewer 1. In particular, we have supplemented the internal reflection function section with explanations of the algorithm (p. 4,§3.3).

---

### Decision · Program_Chairs · 2019-12-19

**Decision:**

Reject

**Comment:**

This paper is very different from most ICLR submissions, and appears to be addressing interesting themes.  However the paper seems poorly written, and generally unclear.  The motivation, task, method and evaluation are all unclear.  I recommend that the authors add explicit definitions, equations, algorithm boxes, and more examples to make their paper clearer.